# Effect of Dexamethasone on Thermoresponsive Behavior of Poly(2-Oxazoline) Diblock Copolymers

**DOI:** 10.3390/polym13091357

**Published:** 2021-04-21

**Authors:** Monika Majerčíková, Peter Nádaždy, Dušan Chorvát, Leonid Satrapinskyy, Helena Valentová, Zuzana Kroneková, Peter Šiffalovič, Juraj Kronek, Anna Zahoranová

**Affiliations:** 1Department for Biomaterials Research, Polymer Institute of the Slovak Academy of Sciences, Dúbravská cesta 9, 845 41 Bratislava, Slovakia; majercikova.monika@savba.sk (M.M.); zuzana.kronekova@savba.sk (Z.K.); 2Institute of Physics of the Slovak Academy of Sciences, Dúbravská cesta 9, 845 11 Bratislava, Slovakia; peter.nadazdy@savba.sk (P.N.); peter.siffalovic@savba.sk (P.Š.); 3International Laser Centre, Department of Biophotonics, Ilkovičova 3, 841 04 Bratislava, Slovakia; Dusan.Chorvat@ilc.sk; 4Department of Experimental Physics, Faculty of Mathematics, Physics and Informatics, Comenius University, Mlynská Dolina, 842 48 Bratislava, Slovakia; leonid.satrapinskyy@fmph.uniba.sk; 5Faculty of Mathematics and Physics, Charles University, Ke Karlovu 3, 121 16 Prague 2, Czech Republic; helena.valentova@mff.cuni.cz; 6Centre for Advanced Materials Application, Slovak Academy of Sciences, Dúbravská cesta 9, 845 11 Bratislava, Slovakia; 7Institute of Applied Synthetic Chemistry, Vienna University of Technology, Getreidemarkt 9/163MC, A-1060 Vienna, Austria

**Keywords:** ring-opening polymerization, self-assembly, stimuli-responsive polymers, drug delivery systems, crystallization

## Abstract

Thermoresponsive polymers play an important role in designing drug delivery systems for biomedical applications. In this contribution, the effect of encapsulated hydrophobic drug dexamethasone on thermoresponsive behavior of diblock copolymers was studied. A small series of diblock copoly(2-oxazoline)s was prepared by combining thermoresponsive 2-*n*-propyl-2-oxazoline (*n*PrOx) and hydrophilic 2-methyl-2-oxazoline (MeOx) in two ratios and two polymer chain lengths. The addition of dexamethasone affected the thermoresponsive behavior of one of the copolymers, *n*PrOx_20_-MeOx_180_, in the aqueous medium by shifting the cloud point temperature to lower values. In addition, the formation of microparticles containing dexamethasone was observed during the heating of the samples. The morphology and number of microparticles were affected by the structure and concentration of copolymer, the drug concentration, and the temperature. The crystalline nature of formed microparticles was confirmed by polarized light microscopy, confocal Raman microscopy, and wide-angle X-ray scattering. The results demonstrate the importance of studying drug/polymer interactions for the future development of thermoresponsive drug carriers.

## 1. Introduction

Stimuli-responsive polymers are able to respond either to internal stimuli, such as pH, temperature, enzyme, and metabolite concentration, or to external stimuli, such as light, heat, electric, magnetic field, and ultrasound [1]. Such polymers are currently in demand particularly in the biomedical field, e.g., controlled drug release, to enhance the therapeutic effect without damaging healthy cells. Biomedical applications of stimuli-responsive polymers are advantageous due to their noninvasiveness and the possibility to trigger on/off release [2].

Thermoresponsive polymers represent one of the most widely studied classes of stimuli-responsive polymers [3]. These polymers exhibit phase separation in solutions with an increase or decrease in temperature. In the case of polymers exhibiting lower critical solution temperature (*LCST*) behavior, the polymer is soluble in the studied solvent at low temperatures. Above a transition temperature, the polymer chain dehydrates and two immiscible liquid phases are formed [4]. This temperature is also referred to as cloud-point temperature (*T_cp_*) since the polymer solution usually becomes turbid at increased temperature. *T_cp_* depends on the polymer concentration in solution, while the minimum of the polymer concentration-*T_cp_* diagram is called *LCST*. Thermoresponsive (co)polymers retain various architectures, such as linear homopolymers (e.g., poly(N-acrylamide)s [5], poly(2-alkyl-oxazoline)s [6], poly(alkylene oxides)) [7], statistical copolymers [8], block copolymers (e.g., poloxamers) [9,10], or graft copolymers (e.g., oligo(ethylene glycol) based graft copolymers) [11].

Poly(2-oxazoline)s (POx) are a particularly interesting class of biomedical-grade polymers due to lack of cytotoxicity, low immunogenicity, and stealth behavior [12,13,14]. They are accessible via living cationic polymerization providing excellent control over molar mass, polymer architecture, and functionalization in both end-groups or side chains [15,16,17]. They are intensively studied for potential use in various biomedical applications, such as drug and gene carriers [18,19], antifouling surfaces [20], hydrogels [21,22], or 3D bioprinting [23], among others. Although ultrasound [24]- and magnetic field [25]- responsive POx-based materials have already been described, the research focuses mainly on thermoresponsive POx. The thermoresponsive behavior of POx depends on the chemical structure of the side alkyl chain. While poly(2-methyl-2-oxazoline)(PMeOx) is water-soluble in the whole temperature range, the hydrophobicity of the polymer increases with the increasing length of the side chain, leading to *LCST* behavior. Typical homopolymers exhibiting thermoresponsive behavior are poly(2-ethyl-2-oxazoline) (PEtOx), poly(2-isopropyl-2-oxazoline) (P*i*PrOx), poly(2-*n*-propyl-2-oxazoline) (P*n*PrOx) and poly(2-cyclopropyl-2-oxazoline). POx possessing the side alkyl chain longer than three carbons are insoluble in water in the whole temperature range [26].

In addition, the *T_cp_* of POx can be predesigned by copolymerization of different 2-oxazoline monomers [17]. A plethora of statistical [27], block [28], and gradient [29] thermoresponsive copolymers have been prepared to adjust the *T_cp_* of resulting materials finely. Moreover, more complex architectures, such as star-shaped copolymers, graft copolymers, and bottle-brush brushes, have been developed [30,31].

The amphiphilic block and gradient copoly(2-oxazoline)s can self-assemble in solution into micelles, which can be used for the solubilization of the hydrophobic drugs. Encapsulation of drugs in polymeric micelles dramatically increases drug concentration in an aqueous environment. It was shown that POx micelles could encapsulate unprecedentedly large amounts (loading capacity up to 50 wt%, aqueous solubility up to 40 g·L^−1^) of hydrophobic drugs [18].

Temperature can potentially be exploited as a stimulus to trigger the release of the drug from polymeric micelles, which provides additional control over the release kinetics of the drug. Two examples of such systems are following: (i) copolymers comprising thermoresponsive block and hydrophilic block, forming micelles above *T_cp_* from the previously fully soluble system, and (ii) copolymers comprising hydrophobic and thermoresponsive block, which leads to the collapse of micellar corona and formation of a precipitate above *T_cp_*. [32] Both approaches are used in drug/gene delivery systems.

Hruby et al. used triblock copoly(2-oxazoline) with thermoresponsive core and hydrophilic corona to design micelles for radionuclide delivery [33]. The thermoresponsive behavior was exploited to trigger self-assembly to form micelles by heating. Sano et al. presented a similar approach for radionuclide delivery [34]. The authors used statistical copolymer of P*i*PrOx and PEtOx. Upon injection into mice, the copolymer self-aggregated to form depots containing radionuclide in tumors. Triblock terpolymer containing a thermoresponsive part, PEtOx-b-P*n*PrOx-b-poly(l-lysine), was also proposed by Kim et al. for the delivery of antisense oligonucleotide [35]. The thermoresponsive part served as a stabilizing interlayer between the charged core and the hydrophilic corona. An unusual application of thermoresponsive polymers was presented by Tiller et al. [36], where thermoresponsive POx containing iminodiacetic acid end groups (POx-IDA) were employed as reversible enzyme activity thermal switches of horseradish peroxidase and laccase. At the temperature below *T_cp_*, they were able to inhibit the activity of horseradish peroxidase and laccase by more than 99%. Increasing the temperature led to POx-IDA precipitation and 100% recovery of the enzyme activity. Thermoresponsive POx were also studied as vectors for transfection of nucleic acids. Mees et al. [37] showed that statistical copolymers of *n*PrOx and poly(ethylene imine) exhibited temperature-induced self-assembly.

In this paper, we extended the work on thermoresponsive POx with the aim to study the thermoresponsive behavior of amphiphilic block POx with and without the encapsulated model drug. Similarly to Hruby et al. [33], we prepared copolymers containing hydrophilic and thermoresponsive parts, to serve as a corona and a core of self-assembled micelles, respectively. Here, hydrophilic MeOx and thermosensitive *n*PrOx with two ratios (9:1 and 8:2) and two polymer chain lengths (DP 100 and 200) have been used. A hydrophobic model drug, dexamethasone, was encapsulated into micelles by a thin-film hydration method. Dexamethasone is an anti-inflammatory corticosteroid drug reported to reduce mortality of COVID-19 patients on mechanical ventilation [38]. Our motivation was to study the effect of the encapsulated drug on the thermoresponsive behavior of the formulation since certain discrepancies can be found in the scientific literature. On the one hand, a straightforward method to release a drug by cooling from the micelles with a thermoresponsive core was proposed by several authors [32,39]. On the other hand, there is growing evidence that the presence of a drug affects the self-assembly behavior of amphiphilic block copolymers [40,41,42]. With this in mind, we studied the effect of the drug on the thermoresponsive behavior of block coPOx by means of dynamic light scattering (DLS), UV/Vis spectroscopy, and optical microscopy. We observed the formation of the micrometer-scale particles, further characterized by optical microscopy, scanning electron microscopy, Raman microscopy and wide-angle X-ray scattering (WAXS). 

## 2. Materials and Methods

### 2.1. Materials

2-Methyl-2-oxazoline (MeOx), purchased from TCI (Zwijndrecht, Belgium), was dried over KOH for 24 h, distilled over CaH_2_ and stored under argon. 2-*n*-propyl-2-oxazoline (*n*PrOx) was prepared from butyronitrile and 2-aminoethanol [43,44], dried over KOH for 24 h, distilled over CaH_2_ and stored under argon. Calcium hydride (CaH_2_), methyl 4-nitrobenzenesulfonate, dexamethasone were purchased from Sigma-Aldrich (Steinheim, Germany) and used as received. Potassium hydroxide (KOH) was purchased from Mikrochem (Bratislava, Slovakia), methanol from Centralchem (Bratislava, Slovakia). Benzonitrile (Sigma-Aldrich, Steinheim, Germany) was distilled over P_2_O_5_ and stored under argon. Diethyleter and ethanol (p.a. 96%) were purchased from Centralchem (Bratislava, Slovakia). Phosphate-buffered saline (PBS) was purchased from Sigma-Aldrich (Dulbecoo´s PBS, Steinheim, Germany).

### 2.2. Polymerizations

Four diblock copolymers (MeOx_n_-b-*n*PrOx_m_) were prepared from MeOx and *n*PrOx, in two different ratios of blocks (0.2:1 and 0.1:1) and in two polymer chain lengths (DP = 100 and 200). Copolymers were synthesized through the living cationic ring-opening polymerization. Copolymer MeOx_80_-b-*n*PrOx_20_ (P1) was prepared as follows: a Schlenk flask with initiator methyl 4-nitrobenzenesulfonate (MeONs, 72 mg, 0.33 mmol) was dried for 30 min under vacuum. The reaction flask with the initiator was transferred to the glove box where the first monomer MeOx (2.53 g, 29.7 mmol) was added with benzonitrile (9 mL). The polymerization of the first block was carried out at 100 °C for 22 h in an oil bath. Subsequently, the second monomer _n_PrOx (0.79 g, 6.98 mmol) and benzonitrile (5.2 mL) were added, the reaction ran for 25 h at 100 °C in an oil bath to complete conversion, as verified by ATR-FTIR measurement. Next, 1M methanolic KOH (0.9 mL) was added and the reaction was terminated over 3 h at RT. The resulting block copolymer was precipitated into cold diethyl ether (200 mL), dialyzed against water for 72 h (SpectraPor6, MWCO of 1kDa, Spectrum Laboratories, Inc., USA) and freeze-dried. The resulting product was obtained as a slightly yellow powder (yield 2.4 g, 72%).

MeOx_90_-b-*n*PrOx_10_ (P2) was prepared similarly using MeONs (59 mg, 0.27 mmol), 2.72 g (31.9 mmol) of MeOx in 10 mL of benzonitrile for the first block and 0.223 g (1.97 mmol) of nPrOx in 1 mL of benzonitrile for the second block. The final product was obtained as a slightly yellow powder (yield 2.16 g, 73%).

MeOx_160_-b-*n*PrOx_40_ (P3) was prepared using (73 mg 0.34 mmol) of MeONs. 4.7 g (55.2 mmol) of MeOx in 12 mL of benzonitrile and 1.47 g (13.0 mmol) of *n*PrOx in 10 mL of benzonitrile was used for the first and second block, respectively. The product was obtained as a white powder (yield 5 g, 81%).

MeOx_180_-b-*n*PrOx_20_ (P4) was prepared using (36 mg 0.17 mmol) of MeONs. 2.73 g (32.1 mmol) of MeOx in 10 mL of benzonitrile and 0.47 g (4.2 mmol) of *n*PrOx in 1.2 mL of benzonitrile was used for the first and second block, respectively. The final product was obtained as a white powder (yield 2.71 g, 85%).

### 2.3. Encapsulation of Dexamethasone (Dexa)

The encapsulation of Dexa was carried out by the thin film hydration method, according to the literature [45]. The stock solutions of Dexa (10 mg mL^−1^) and the copolymer (100 mg mL^−1^) in ethanol were prepared. Subsequently, the stock ethanolic solutions of both components were mixed in desired ratios. After mixing, the ethanol was evaporated by heating using a hot-air gun, and a dry, thin polymeric drug film was formed on the walls of glass vials. The thin film was dissolved in 1 mL of PBS or distilled water at RT and redispersed by Vortex. The concentration of copolymer in the resulting solution was kept to 10 mg mL^−1^; the concentrations of Dexa were 0.1, 0.5, 1, and 2 mg mL^−1^.

### 2.4. NMR Spectroscopy

NMR spectroscopy was used to determine the final composition of prepared block copolymers P1–P4. ^1^H NMR spectra were measured on an instrument Varian VXR-400 (Varian, Wilmington, DE, USA) in CDCl_3_ at room temperature using tetramethylsilane as an internal standard.

### 2.5. GPC Measurements

Instrumentation used in GPC characterization consisted of the pumping system type P102 (Watrex, Prague, Czech Republic) and evaporative light scattering detector ELS—1000 (PL-Agilent Technologies, Stretton, UK). The temperature of evaporators was set to 180 °C, and the gas flow rate was 1.5 mL min^−1^. GPC column was TSKgel B0076 from Tosoh Bioscience (Japan). The flow rate was set to 1 mL min^−1^. The GPC measurements were performed at room temperature. The mixture of 50 wt.% *N*,*N*-dimethylformamide, HPLC grade > 99.7% from Alfa Aesar and 50 wt.% chloroform, HPLC grade ≥ 99.8%, amylene stabilized from Sigma-Aldrich (Germany) was used as GPC eluent. PMMA was used as a calibration standard. Data were collected and processed with the help of Clarity software (DataApex, Czech Republic).

### 2.6. UV/Vis Spectrophotometry

A UV-1800 instrument (Shimadzu, Kyoto, Japan) equipped with a six-cell thermoelectrical temperature controller CPS-240A (Shimadzu, Kyoto Japan) was used for measuring the UV/Vis spectra. The samples P1–P4 (10 mg mL^−1^ in PBS) without and with encapsulated Dexa (0.1, 0.5, 1, and 2 mg mL^−1^) were freshly prepared by a thin-film method prior to measurement. The measurement was performed in the temperature range of 10–40 °C with a 3 °C step. The equilibration time was 5 min. The transmittance at a wavelength of 500 nm was evaluated.

### 2.7. Dynamic Light Scattering (DLS)

DLS was used to measure particle size in a P1 and P2 copolymer solution (10 mg mL^−1^) in PBS without or with the addition of Dexa (0.1, 0.5 mg mL^−1^). A Zetasizer Nano-ZS instrument (Malvern Instruments, Malvern, UK) with a 4 mW helium/neon laser (λ = 633 nm) was used for measurement. The behavior of the solutions in the temperature range of 10–40 °C with a 3 °C temperature step and the equilibration time of 5 min was measured. Finally, the samples were cooled to 10 °C after reaching 40 °C. The scattered light intensity was measured at an angle of 173°. The results (hydrodynamic diameter, D_h_) are displayed as a peak maximum (mode) from intensity-weighted or volume-weighted size distribution, as indicated.

### 2.8. Optical Microscopy

An inverse optical microscope (Optika Microscopes, Italy, Axiovert 200M, Zeiss, Jena, Germany) equipped with a heating plate was used to observe the formation of microparticles. We used LD Plan-Neofluar 20×/0.4 objective with temperature-stabilized sample holder and temperature control unit TempControl 37-1. Copolymer P1 at 10mg·mL^−1^ without Dexa and with 0.5, 1, and 2 mg·mL^−1^ Dexa was examined. The samples were prepared by thin-film hydration method in PBS freshly prior to the measurement. One hundred microliters of the sample was pipetted onto a glass slide, covered with a petri dish to avoid evaporation during measurement. The samples were measured in a temperature range 25–40 °C, with a heating step 3 °C and equilibration time of 10 min. The number of microparticles was calculated using an open-source imaging processing software (Image J from National Institutes of Health, Bethesda, Maryland, USA).

The anisotropic texture of microparticles was determined by a polarizing optical microscope Nikon Eclipse 80i equipped with a heating stage (Linkam LTS350, Linkam Scientific Instruments Ltd., Tadworth, UK). Copolymer P1 at 10 mg·mL^−1^ and 2 mg·mL^−1^ Dexa prepared by the thin-film method in PBS was examined at RT to prove the crystalline structure of the microparticles. 10–20 μL of the sample were pipetted onto a glass slide, covered with a cover glass to avoid evaporation during measurement.

### 2.9. Scanning Electron Microscopy (SEM)

P1 microparticles were prepared by the thin-film hydration method with a final concentration of 10 mg·mL^−^1 of copolymer P1 and 2 mg·mL^−1^ of Dexa dissolved in PBS. The microparticles were subsequently air-dried on a Si wafer. The morphology of microparticles was examined using microscope Lyra 3 (Tescan, Brno, Czech Republic), operating at 5 kV. The chamber pressure was 3.2 × 10^−2^ Pa. The SEM images were obtained using secondary electrons.

### 2.10. Wide-Angle X-ray Scattering (WAXS)

WAXS measurements were performed on a custom-designed set-up (Nanostar, Bruker AXS, Karlsruhe, Germany) equipped with a liquid-metal-jet Ga microfocus source (Excillum, Kista, Sweden) and parallel Montel optics (Incoatec, Geesthacht, Germany). Beam collimation was done by two 550 µm scatterless Ge pinholes (Scatex, Incoatec, Geesthacht, Germany) at a distance of 50 cm from each other. Pinholes and the sample were placed in an evacuated chamber with a large beryllium X-ray window and a two-dimensional hybrid pixel detector (Dectris, Pilatus 300K, Baden-Daettwil, Switzerland) in close proximity behind it. Sample positioning was maintained by vacuum-compatible hexapod (Physik Instrumente, Karlsruhe, Germany). Freeze-dried P1 copolymer powder and Dexa powder were measured in WAXS geometry with an exposure time of 1000 s. P1 microparticles were prepared by the thin-film hydration method with a final concentration of 10 mg·mL^−1^copolymer P1 and 2 mg·mL^−1^ Dexa dissolved in distilled water. A drop of the microparticle sample was pipetted on a Si wafer and air-dried at room temperature. P1 microparticles were measured in grazing incidence WAXS (GIWAXS) with an angle of incidence of 0.2°. The measured data are presented in the form of reciprocal maps. Presented 1D profiles were obtained from the cuts along the q_z_ scattering wave vector and were subsequently recalculated to the angle (2θ) for Cu Kα radiation (λ = 1.54 Å).

### 2.11. Confocal Raman Microscopy (CRM)

CRM was performed using the WITec alpha 300 R+ confocal Raman microscope equipped with the WITec UHTS300 spectrometer. The sample was prepared by a thin-film hydration method with a concentration of 10 mg·mL^−1^ of copolymer P1 and 1 mg·mL^−1^ of Dexa in distilled water and measured in a hydrated state. Raman signal was acquired after excitation with the laser at 785 nm, water immersion objective Carl Zeiss 20×/1NA (Carl Zeiss, Jena, Germany), at a controlled room temperature of 23 °C. Raman spectra were analyzed using Project Four+ software from WITec GmbH. Polymer and Dexa powders spectra were analyzed to identify specific vibration modes for Dexa. Dexamethasone amorphous and crystalline phase was distinguished as described in ref [46]. Raman spectrum of Dexa in an amorphous phase was obtained from ethanolic Dexa solution (1 mg·mL^−1^). Distribution of P1 and Dexa within the particle was performed by Project Four+ software (Witec, Germany) using mathematical Gauss fit of the area under the peak at the positions 1490 rel.·cm^−1^ for polymer and 1660 rel.·cm^−1^ for Dexa.

## 3. Results and Discussion

### 3.1. Synthesis and Characterization of Diblock Copolymers

Four diblock copoly(2-oxazoline)s differing in molar masses and monomer ratio were synthesized using cationic ring-opening polymerization (Figure 1a). Copolymers with two polymer chain lengths of 100 and 200 monomer units and two different molar ratios of hydrophilic MeOx and thermoresponsive *n*PrOx (9:1, 8:2) were prepared. *n*PrOx was selected based on our previous results where we demonstrated a relatively high loading capacity of similar copolymers for the model drug dexamethasone (Figure 1b), compared to other copolymers [24]. The synthesized polymers were characterized by ^1^H NMR spectroscopy and GPC, and the results are shown in Table 1. ^1^H NMR spectra are shown in Appendix A. The experimental ratios of the blocks corresponded well with the theoretical ratios of the monomers in the feed. The DP (degree of polymerizations) calculated from ^1^H NMR were higher compared to the values from GPC, which can be attributed to the detection limit of the method. Molar masses obtained from GPC were lower compared to the theoretical values. However, the values should be considered as relative since amphiphilic block copolymers containing two blocks of different polarity were analyzed using polymethylmethacrylate (PMMA) as a standard. The dispersities of the prepared copolymers were higher than the values typical for living cationic (co)polymerization of 2-oxazolines, which may be due to lower solubility of longer PMeOx polymers in the polymerization solvent benzonitrile, leading to a less effective attachment of the second block. It should be noted that although POx is considered to be nonbiodegradable in vivo by hydrolytic and enzymatic pathways, the cut-off for glomerular filtration for pEtOx was found around 40 kDa [47], which is above molar masses used in this study.

### 3.2. Thermoresponsive Behavior of Block Copolymers in PBS

First, we examined the thermoresponsive behavior of four studied diblock copolymers P1–P4 without or with the addition of different amounts of dexamethasone (Dexa) (0.1, 0.5, 1, and 2 mg·mL^−1^). P*n*PrOx homopolymer exhibits *LCST* behavior, manifested as a sharp decrease of transmittance above the *T_cp_*, which was reported around 26 °C for a polymer with DP 100 at *c* = 10 mg·mL^−1^ [27]. Therefore, we measured transmittance of the aqueous solutions at wavelength 500 nm of P1- P4 copolymers at 10 mg·mL^−1^ by UV/Vis spectroscopy in the temperature range from 10 °C to 40 °C. The transmittance as a function of the temperature of copolymer solutions without Dexa is depicted in Figure 2a–d (black squares). The thermoresponsive behavior of the copolymers depended not only on their chemical composition but also on the length of polymer chains. The copolymers P1 and P2, possessing shorter polymer chains (DP 100), exhibited no change in transmittance with increasing temperature; the solutions remained clear throughout the temperature range studied. However, it is impossible to recognize by the UV/Vis measurement whether the copolymers are fully soluble in water or whether they exhibit temperature-induced self-assembly at the nanoscale, as we observed in some cases for ABA triblock copoly(2-oxazoline)s in our previous work [43]. On the other hand, copolymer P3 (*n*PrOx_40_-MeOx_160_) was turbid in the whole studied temperature range. This is an unexpected finding since P*n*PrOx exhibits *T_cp_* around 26 °C (DP 100, c = 10 mg·mL^−1^) [27], and it seems unlikely that the addition of even more hydrophilic MeOx comonomer will lead to a decrease of *T_cp_* or even will make the copolymer insoluble in water. However, the self-assembly of copoly(2-oxazoline)s was shown to depend strongly on the preparation method, as recently demonstrated by Filippov et al. [48] for fluorinated copoly(2-oxazoline)s. This can also be our case since the control copolymer samples without Dexa were prepared analogously to the samples with Dexa, i.e., by thin-film hydration method. The only copolymer exhibiting “typical” thermoresponsive behavior, i.e., transmittance change of the sample solutions, was copolymer P4 (*n*PrOx_20_-MeOx_180_). Without the addition of Dexa, the aqueous solution of copolymer showed *T_cp_* at 30 °C, which is close to a value to *T_cp_* of homopolymer P*n*PrOx at 10 mg·mL^−1^. Overall, the results show that both chain length and the ratio between the blocks influence the water-solubility and thermoresponsive behavior of the studied copolymers.

Further, we studied the effect of encapsulated Dexa on the thermoresponsive behavior of the samples. The thermoresponsive behavior of solutions of samples containing four different concentrations of Dexa, 0.1, 0.5, 1, and 2 mg·mL^−1^ and 10 mg·mL^−1^ of copolymers P1-P4 in PBS was also studied by UV/Vis spectroscopy. These concentrations were selected based on our previous study on loading of Dexa into POx-based micelles [24]. The UV/Vis spectra of P1-P4 with various concentrations of Dexa are depicted in Figure 2a–d. The effect of the encapsulated Dexa on thermosensitive behavior was evident in the case of P4 (*n*Pr_20_MeOx_180_), the only copolymer exhibiting clouding behavior even without the drug. Already after the addition of 0.1 mg·mL^−1^, *T_cp_* was shifted slightly to lower values, i.e., from 30 °C to 28.5 °C. With increasing Dexa concentration, *T_cp_* was further decreasing, up to 12 °C for 1 mg·mL^−1^ of Dexa. The copolymer solution containing 2 mg·mL^−1^ of Dexa was turbid in the whole studied temperature range. This finding has an implication for the development of thermoresponsive polymeric drug carriers, where *T_cp_* of the copolymer is precisely matched to the human body temperature, but the effect of the added drug is usually not considered. As an opposite example, Uchman et al. [49] studied the effect of encapsulated metallacarborane cobalt bis(dicarbollide) anion (COSAN), potentially applicable in cancer and AIDS treatment, on thermoresponsive behavior of similar diblock copolymer P*n*PrOx_80_-PMeOx_40_. Similarly to our results, the authors described the shift of T_cp_ of the system to lower values upon the addition of lower amounts of COSAN. The addition of higher amounts of COSAN led to the disappearance of *LCST* behavior of the sample; the COSAN-polymer complexes remained in the form of desolvated nanoparticles.

For the other copolymer samples P1–P3, the addition of lower amounts of Dexa did not affect the themoresponsive behavior of the samples and the copolymers remain clear or turbid over the entire temperature range investigated. However, at higher concentrations of Dexa, we observed unusual heating curves, with regions of increasing and decreasing transmittance values. We were unable to fit such curves by Boltzmann function, opposite to “typical” thermoresponsive behavior. This observation indicates the occurrence of several processes during the heating of the samples. The formation of visible large aggregates, which precipitated during the measurement, led to an apparent increase of transmittance values.

To shed light on this behavior, we decided to examine the morphology of the formed precipitate under an optical microscope. The optical images of the aqueous solutions of 10 mg·mL^−1^ of P1–P4 without Dexa and with 2 mg·mL^−1^ Dexa, measured at RT after the UV/Vis measurements, are shown in Figure 2. The copolymer solutions of P1, P2, and P4 without Dexa were clear and no precipitate was visible, consistent with high transmittance values in UV/Vis measurement. However, tiny droplets were seen on microscopic images of sample P3, corresponding to turbid solution as measured by UV/Vis spectrometer. Similar droplets formed in heated solutions of star-shaped POx copolymers were described by Sato et al. [50]. On the other hand, the copolymer solutions with 2 mg·mL^−1^ Dexa contained separate microparticles and aggregates of microparticles, as shown in Figure 2. To the best of our knowledge, the formation of similar microparticles has not yet been described for any POx-drug combination, including our block copolymers with encapsulated Dexa [24]. In that case, we used diblock copolymers with a longer P*n*PrOx block. It should also be emphasized that the control sample containing Dexa without any copolymer, prepared by the thin-film method, i.e., dissolved in ethanol, dried and redissolved in PBS, contains Dexa crystals possessing different size and shape (see Appendix A), which leads us to the assumption that the formation of microparticles is influenced by the presence of POx copolymers.

Further, we examined in closer detail solutions of P1 and P2 at 10 mg·mL^−1^ in PBS with 0, 0.1, and 0.5 mg·mL^−1^ of Dexa by dynamic light scattering (DLS) measurement since these samples remained clear through the whole examined temperature range. We hypothesized that either the copolymer chains remained soluble also upon heating (in a coil conformation) or the temperature-induced phase transitions resulted in a formation of nanosized particles, which did not lead to turbidity change. The results from DLS measurements, presented as intensity-weighted hydrodynamic diameter *D_h_* at 10 °C and 40 °C, are displayed in Figure 3a–e. Copolymer solutions without Dexa (Figure 3a,d) exhibited the presence of soluble copolymer coils (*D_h_* < 10 nm), together with a fraction of bigger aggregates (*D_h_* > 10 nm). In the case of P2, this size distribution of bigger aggregates was bimodal, which may indicate the presence of two distinct populations of scattering objects but also the presence of nonspherical aggregates. These two cases are indistinguishable by used DLS instrument. However, with the increase of temperature, the size distributions did not change, proving that studied copolymers P1 and P2 did not exhibit *LCST* behavior in the studied concentration and temperature range, probably due to the shorter P*n*PrOx chain. The presence of 0.1 mg·mL^−1^ Dexa resulted in a more polydisperse size distribution (Figure 3b,e), although even in this case, the smaller peak attributed to copolymer coils also persisted at 40 °C. On the other hand, the sample P1 containing 0.5 mg·mL^−1^ Dexa behaved differently. In this case, the smaller peak (*D_h_* < 10 nm) completely disappeared upon heating, while the intensity-weighted distribution became bimodal, with *D_h_* 33 and 255 nm. A similar observation was reported in our previous work dealing with different triblock copoly(2-oxazoline)s for the samples with higher PMeOx content [43]. Complementarily to intensity-weighted size distribution, we displayed volume-weighted size distribution, as was previously also used by Hruby et al. [33] to characterize thermoresponsive poly(2-oxazoline)s. The comparison of intensity- and volume-weighted size distribution for sample P2 at the temperature of 40 °C is shown in Appendix A. The bigger particles (*D_h_* > 10 nm) were not visible in the volume-weighted distribution since they represented only a minor volume fraction of the particles in solution. Similarly, the volume-weighted distributions of all polymer samples were compared in Figure 3f. Although the samples P1 and P2 without Dexa and with 0.1 mg·mL^−1^ Dexa maintained the major fraction of small particles (*D_h_* < 10 nm) throughout the whole temperature range studied, sample P1 with 0.5 mg·mL^−1^ Dexa exhibited sudden change at 31 °C when the bigger aggregates with a diameter around 300 nm prevailed. This process was reversible, as the aggregates dissociated upon cooling (see Appendix A, representing sample P1 with 0.5 mg·mL^−1^ at various temperatures), implying a potential use for controlled drug delivery. However, as discussed further in the following sections, we observed a formation of microparticles (diameter ≈ 10 μm) upon heating also in this sample, although not detectable via UV/Vis and DLS, possibly due to the sedimentation. In this case, the intravenous administration of such drug formulation would be hampered.

After detailed characterization of thermoresponsive behavior of copolymer solutions in PBS by UV/vis spectrometry and DLS, we decided to examine in detail the formation of the microparticles with an optical microscope. We selected copolymer P1 as a model system for further detailed examination since it displayed the most irregular heating curves (see Figure 2).

### 3.3. Formation of Microparticles in P1 Copolymer/Dexa Solutions

The effect of concentration of Dexa and temperature on the formation of microparticles was studied. The solutions of P1 in PBS were observed under an optical microscope equipped with a heating plate. The formation of the microparticles in a drop of sample solutions of different Dexa concentrations (10 mg·mL^−1^ P1 sample; 0, 0.5, 1, and 2 mg·mL^−1^ of Dexa in PBS) was investigated. The samples were freshly prepared prior to measurement. The formation of microparticles with increased temperature was not observed for the control samples of polymer solution without Dexa (Appendix A).

The number and size of the microparticles were evaluated using ImageJ software. The distribution of microparticle size for temperatures 25 °C and 37 °C for three different Dexa concentrations is compared in Figure 4. Optical images of microparticles at 25 °C are shown in Appendix A. At 25 °C for the lowest Dexa concentration (0.5 mg·mL^−1^ ), almost no particles were formed. With increasing the Dexa concentration to 1 mg·mL^−1^, a small number of smaller microparticles (<20 μm) was formed. For the highest examined concentration of Dexa, 2 mg·mL^−1^, both small (<20 μm) and larger microparticles (>20 μm) were present already at 25 °C. The situation was different at 37 °C, where the number of microparticles increased for all three concentrations compared to 25 °C. However, the size and morphology of the formed microparticles differed for different concentrations of Dexa. While at 0.5 mg·mL^−1^ Dexa, only smaller elongated microparticles (<20 μm) with a few nonspherical larger (>20 μm) microparticles were formed, at 1 mg·mL^−1^, almost exclusively larger, spherical particles were present. In the case of 2 mg·mL^−1^ Dexa, both larger spherical particles (>20 μm) and smaller (<20 μm) elongated particles were present in high amounts.

The dependence of the number of microparticles on temperature ranging from 25 °C to 40 °C is shown in Figure 5. For the sake of clarity, the particles were arbitrarily divided into two categories, smaller particles (<20 μm) and larger particles (>20 μm). For the highest Dexa concentration (2 mg·mL^−1^, Figure 5c), a high number of both smaller and larger particles could be seen at 25 °C, and only a minor increase in the number of smaller particles was observed at 28 °C. On the other hand, for 0.5 mg·mL^−1^ Dexa (Figure 5a), both small and larger particles appear at 31 °C and their number gradually increases with increasing the temperature. In the case of 1 mg·mL^−1^ of Dexa (Figure 5b), the number of smaller particles remained constant in the whole temperature range, while the larger particles appeared at 31 °C. Above the temperature of 34 °C, the number of larger particles for 1 mg·mL^−1^ of Dexa remained constant.

When the sample solutions were cooled down to 8 °C after measurement and incubated for 15 min, the microparticles were still present in the solutions (Appendix A).

### 3.4. Formation of Microparticles in P1 Copolymer/Dexa Solutions

To characterize the formed microparticles in more detail, we also employed polarized-light microscopy. Since the formation of microparticles was irreversible (we did not observe dissolution of the formed structures upon cooling), we assumed that the crystallization of the drug could take place. The images from the polarized-light microscope of the sample P1 10 mg·mL^−1^ and Dexa 2 mg·mL^−1^ in distilled water at RT are depicted in Figure 6. Regular, crystalline structures appeared bright using an optical microscope with polarized light. In our case, it can be seen that the encapsulated Dexa/P1 copolymer microparticles form crystalline, regular structures. In Figure 6, the same particle position for visible and polarized light is compared to emphasize the crystallinity of the sample. It should be noted that, in this case, a drop of solution with Dexa encapsulated in P1 was applied between the glass slides to allow the measurement, as the measurement was performed in a closed chamber connected to liquid nitrogen for cooling and on a heating pad. This procedure was slightly different from the measurement with an optical microscope discussed in the previous section, where the liquid drop of the sample solution was not covered by a glass slide. In both cases, the shape and size of microparticles remained unchanged. However, slight differences in microparticle formation kinetics were observed. Similar morphologies of crystalline structures were observed in a study of biomimetic crystallization of calcium carbonate in the presence of poly(ethylene glycol)-b-poly(methacrylic acid) [51]. The used block copolymer possessing a double-hydrophilic structure acted as a crystal modifier. Interestingly, the copolymer P1 used in our case also had only a very short P*n*PrOx chain, and it did not exhibit any transmittance changes after heating up to 40 °C. It can thus be considered as a double-hydrophilic copolymer. A similar study on controlled crystallization was not performed with dexamethasone to the best of our knowledge. However, it was shown that when dexamethasone is combined with poly(dodecyl methacrylate), the polymer may act as a nucleation site to form a crystalline drug form [46]. It should be noted that the study [46] was performed on thin films of polymer/drug mixtures.

The morphology of the microparticles from P1 copolymer (10 mg·mL^−1^) and Dexa (2 mg·mL^−1^) in a dried state was studied by scanning electron microscopy (SEM) and wide-angle X-ray scattering (WAXS) methods (Figure 7a–c). The detailed microstructure of air-dried microparticles obtained from SEM is shown in Figure 7c. It could be seen that the microparticle retained a similar morphology as in a hydrated state even after drying. Moreover, the SEM images indicated that the middle part of the microparticle is deepened. 

Dexa and copolymer P1 were measured separately in solid-state using WAXS geometry. The results are shown in Figure 7a,b. Dexa diffraction pattern reveals high crystallinity of the sample. The characteristic diffraction peaks appear at 2θ = 14.2, 16.8, and 18.4°. On the other hand, the absence of Debye–Scherrer rings in the P1 copolymer powder diffraction pattern suggests no crystalline phase. This finding is not surprising since both PMeOx and P*n*PrOx are generally regarded as amorphous polymers [52]. Due to the large thickness of the sample, P1 microparticles were measured in grazing incidence WAXS (GIWAXS) with an angle of incidence of 0.2° Debye–Scherrer rings are visible, suggesting that Dexa remains in the crystalline phase. The extracted 1D diffraction profiles reveal similar characteristic peaks as in the case of Dexa powder (2θ = 14.2, 16.8, and 18.4°). It should be noted that the intensities of the peaks could not be compared quantitatively due to the different measurement set-up and different sample forms (Dexa powder vs. dried microparticles). In summary, GIWAXS measurement of the samples in the dried state confirmed that the microparticles were composed of Dexa in the crystalline phase, corroborating well with the polarized light microscopy results.

Microparticles in the hydrated state from P1 (10 mg·mL^−1^) and Dexa (1 mg·mL^−1^) were also studied by confocal Raman microscopy (CRM). The spectra of Dexa in amorphous (ethanolic solution of Dexa, 1 mg·mL^−1^) and crystalline form differ only subtly in the region of 1600–1620 rel. cm^−1^. In the amorphous phase, solely one broad vibration mode is observed. However, in crystalline form, the vibration mode is clearly divided into two separate peaks (Appendix A).

Raman spectra of P1 and Dexa are compared in Figure 8 where some peaks specific for each component can be recognized in both spectra. However, in Raman spectra of particles, the signal intensity for P1 was very low compared to the signal of Dexa. The specific peaks of P1 at the positions 1044, 1490, and 1636 rel. cm^−1^ were very weak but recognizable in the spectrum of a particle, as indicated by red arrows in Figure 8a. From the microparticle spectrum, it was also clear that the drug is in the crystalline form as the two separate peaks in the region 1600–1620 rel.·cm^−1^ were present (Figure 8a). The P1 and Dexa signals were recognizable within the whole microparticle, and the spatial distribution of both components within the two different particles is shown in Figure 8b,c. The middle part of the microparticle profile was deepened similarly to the SEM image.

## 4. Conclusions

In this paper, we studied the effect of added Dexa on the thermoresponsive behavior of diblock copoly(2-oxazoline)s. With this aim, we prepared a small series of four diblock copolymers P1–P4 composed of MeOx and *n*PrOx with two different ratios (9:1, 8:2) and two different polymer chain lengths (DP 100 and 200). The thermoresponsive behavior of diblock copolymers in water was found to be dependent on the copolymer composition and the presence of Dexa. In the absence of Dexa, temperature-triggered turbidity was observed for the copolymer P4 (*n*PrOx_20_MeOx_180_), while the solutions of copolymers P1 and P2 with DP 100 remained clear over the entire temperature range investigated. On the other hand, the solution P3 (*n*PrOx_40_MetOx_160_) was turbid also at low temperatures. Upon the addition of Dexa, the T_cp_ of copolymer P4 solution shifted to lower temperature depending on Dexa concentration. In addition, we observed a formation of microparticles in Dexa/copolymer solutions in PBS. The number and size of the formed microparticles depended on the concentration of Dexa and temperature, with the most regular microparticles of the diameter around 30 μm obtained at the concentration of 1 mg·mL^−1^ Dexa and 10 mg·mL^−1^ of P1. The detailed investigation by WAXS, confocal Raman microscopy, and polarized light microscopy revealed a crystalline structure of the microparticles. The formation of microcrystals in the drug solution may affect the in vivo performance of the polymer-drug formulation if the micellar nanosized formulation is intended for intravenous drug administration. On the other hand, micron-sized drug formulations can be used for, e.g., oral drug administration to enhance the dissolution rate of poorly water-soluble drugs [53]. Another potential application of such diblock copoly(2-oxazolines) could be the controlled growth of calcium phosphate [54] or calcium carbonate for biomineralization purposes, but the effect of POx on other crystalline materials requires further examination.

## Figures and Tables

**Figure 1 polymers-13-01357-f001:**
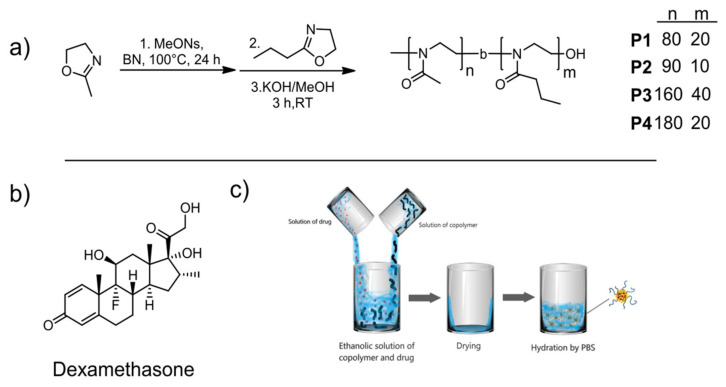
Reaction scheme of the synthesis of diblock copolymers P1–P4 (**a**). Chemical structure of dexamethasone (**b**). Depiction of encapsulation of drug by thin-film method (**c**).

**Figure 2 polymers-13-01357-f002:**
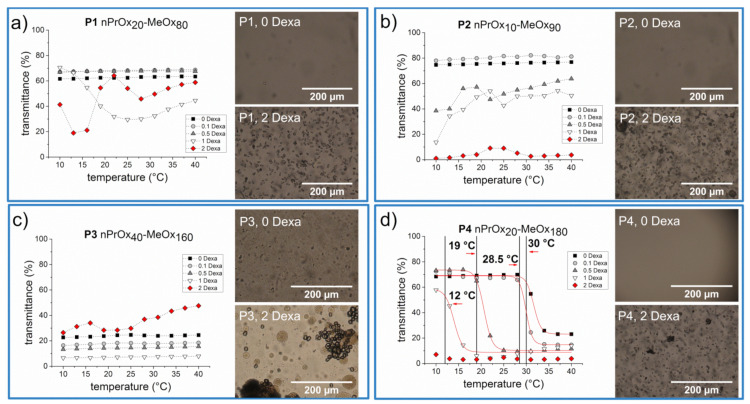
Thermoresponsive behavior of P1 (**a**), P2 (**b**), P3 (**c**), P4 (**d**) copolymers (10 mg·mL^−1^ in PBS) with various concentrations of Dexa (0, 0.1, 0.5, 1, 2 mg·mL^−1^) measured by UV/Vis spectrometry. The red solid line represents Boltzmann fit, T_cp_ values were calculated as 90% transmittance from Boltzmann fits. The black dashed lines serve as visual guidance. Images from the optical microscope represent samples without Dexa and with 2 mg·mL^−1^ Dexa after UV/Vis measurement, captured after cooling to RT.

**Figure 3 polymers-13-01357-f003:**
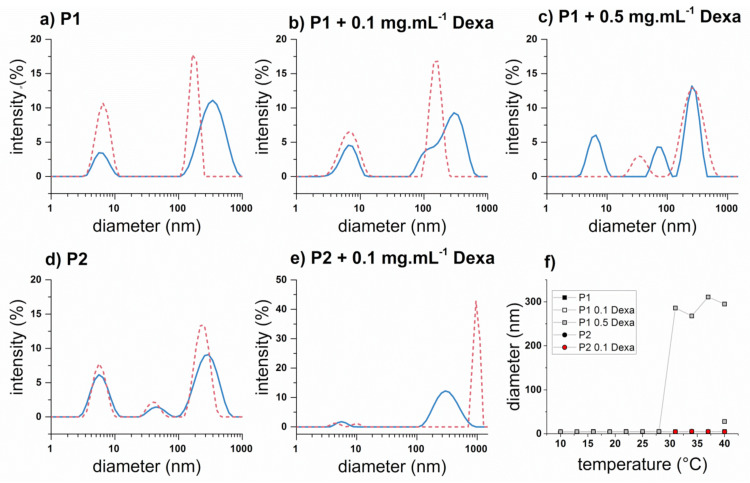
Thermoresponsive behavior of P1 and P2 copolymers (10 mg·mL^−1^ in PBS) with various concentrations of Dexa (0, 0.1, and 0.5 mg·mL^−1^) measured by DLS. Plots (**a**–**e**) represent intensity-weighted size distributions of cold (10 °C, blue solid line) and hot (40 °C, red dashed line) copolymer solutions, plot (**f**) represents *D*_h_ from volume-weighted size distribution as a function of temperature for all samples.

**Figure 4 polymers-13-01357-f004:**
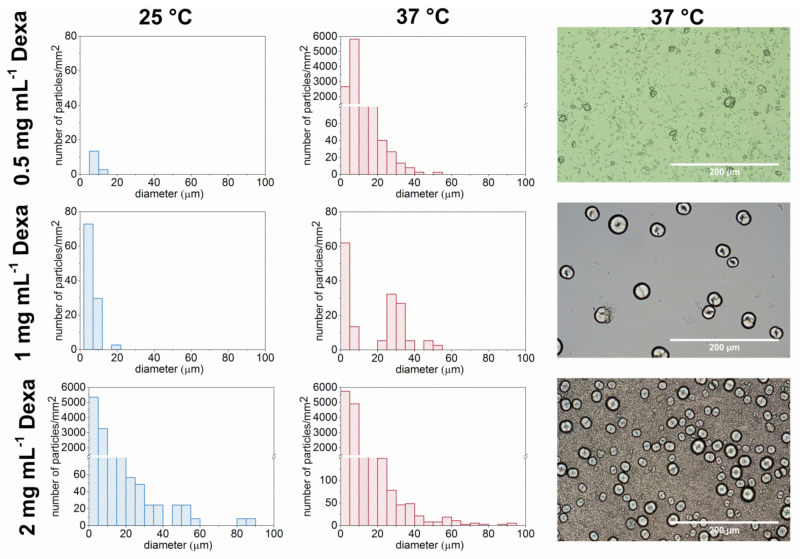
The comparison of morphology and number of microparticles formed from P1 copolymer (10 mg·mL^−1^) and Dexa of various concentrations at 25 °C and 37 °C in PBS. The sample droplet was observed under an optical microscope with a heating plate. The images were evaluated using ImageJ software. The number of the particles per mm^2^ is calculated from three different locations on the sample.

**Figure 5 polymers-13-01357-f005:**
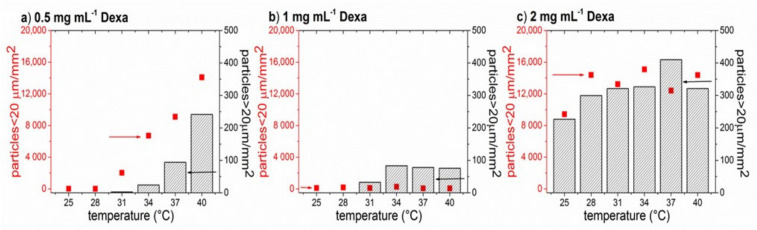
The temperature dependence of the number of smaller (squares) and larger (columns) microparticles formed from P1 copolymer (10 mg·mL^−1^) and various concentrations of Dexa at 25 °C and 37 °C in PBS. The sample droplet was observed under an optical microscope with a heating plate. The images were evaluated using ImageJ software. The density of the particles is calculated from three different locations on the sample.

**Figure 6 polymers-13-01357-f006:**
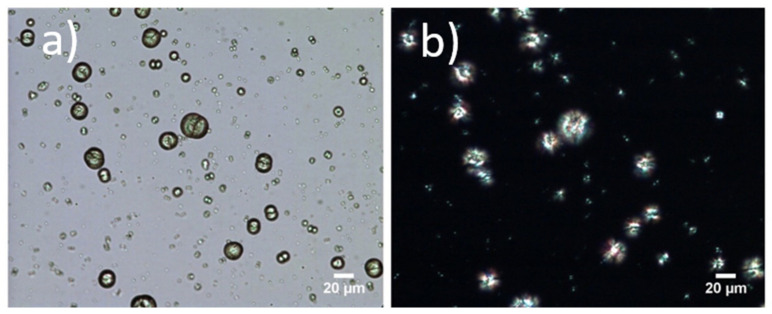
Microparticles formed from 10 mg·mL^−1^ of P1 and 2 mg·mL^−1^ of Dexa in distilled water examined by optical microscopy (**a**) with visible light, (**b**) with polarized light at room temperature.

**Figure 7 polymers-13-01357-f007:**
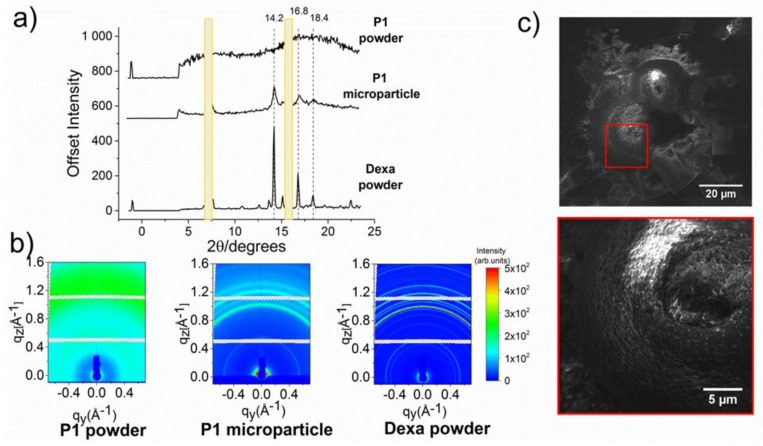
Characterization of dried microparticles from P1 copolymer (10 mg·mL^−1^) and Dexa (2 mg·mL^−1^), examined by WAXS (1D profiles of Dexa powder, polymer P1 powder, and microparticles in (**a**), reciprocal maps in (**b**), SEM images (**c**).

**Figure 8 polymers-13-01357-f008:**
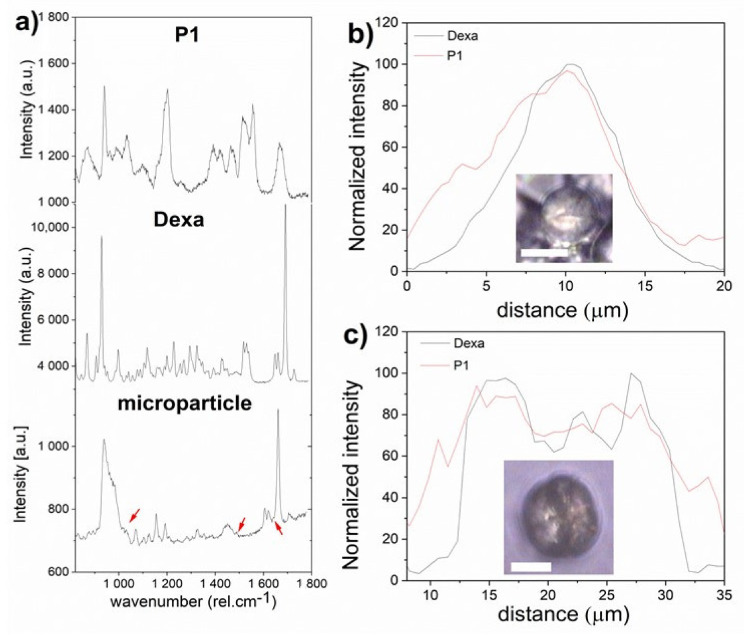
Raman spectra of polymer P1, Dexa, and microparticle are shown (**a**). The red arrows point on peaks corresponding to the polymer within the spectrum of the microparticle. The distribution of polymer and the drug within two particles is shown in (**b**,**c**). The white scale bar in the incorporated images represents 10 µm.

**Table 1 polymers-13-01357-t001:** Molecular characteristics of the prepared diblock copolymers compared to calculated theoretical values.

Sample	DP	Ratio (Theor.) *n*PrOx:MeOx	Mn Theor. (g·mol^−1^)	Mn Exp. (g·mol^−1^) (GPC)	Ratio (NMR) *n*PrOx:MeOx	Đ (GPC)	Yield (%)
P1	100	20:80 (0.25)	9100	9100	0.21	1.55	72
P2	100	10:90 (0.11)	8800	5700	0.11	1.88	73
P3	200	40:160 (0.25)	18,200	11,300	0.22	1.85	81
P4	200	20:180 (0.11)	17,600	10,700	0.13	1.71	85

## Data Availability

The data presented in this study are available in article and Appendix A.

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
