# Peer review of "Effect of Dexamethasone on Thermoresponsive Behavior of Poly(2-Oxazoline) Diblock Copolymers"

_polymers, 2021, doi:10.3390/polym13091357_

Round 1

Reviewer 1 Report

Dear authors, you have submitted a paper that is based on many previous studies. These are block copolymers that are temperature sensitive and are supposed to release an active ingredient (steroid as model). Unfortunately, a lot of similar work has already appeared in this sector. I would accept the work if you consider the following:

- Please explain exactly why thermo reversible polymer block was used in  your work. You could also use block copolymers made of hydrophilic and hydrophobic sequences.

- Please also give a comment whether the polymers are degradable.

Reviewer 2 Report

The authors have investigated thermoresponsive behavior of hydrophobic drug dexamethasone encapsulated with diblock copoly(2-oxazoline)s. The authors have also studied thermoresponsive behavior of dexamethasone with nPrOx20-MeOx180 in the aqueous medium by shifting the cloud point temperature to lower values.

During the heating of the samples, the formation of microparticles containing dexamethasone has been obtained. The authors have demonstrated that the morphology and number of microparticles have affected by structure and concentration of copolymer, concentration of drug, and temperature. The microparticles have been characterized by polarized light microscopy, confocal Raman microscopy and wide-angle X-ray scattering.

This manuscript is a well-written and well-organized paper. Overall, this work can inspire more drug/polymer designs for thermoresponsive drug carriers. Therefore, I would like to recommend this work to publish in Polymers as its current form.

Author Response

We thank the reviewer for the positive feedback. Minor grammar errors were corrected to make the manuscript more readable.

Round 2

Reviewer 1 Report

The revised  paper can be published...